# Invited perspectives: Views of 350 natural hazard community members on key challenges in natural hazards research and the Sustainable Development Goals

Robert Šakić Trogrlić[1,2], Amy Donovan[3], Bruce D Malamud[1]

[1]Department of Geography, King's College London, London, WC2B 4BG, UK.
[2]Systemic Risk and Resilience (SYRR) Group, Advancing Systems Analysis (ASA) Program, International Institute for Applied Systems Analysis (IIASA), Laxenburg, 2361, Austria.
[3]Department of Geography, University of Cambridge, Cambridge, CB2 3EN, UK.

*Correspondence to*: Bruce D. Malamud (bruce.malamud@kcl.ac.uk)

**Abstract.** In this paper, we present the results of an NHESS (*Natural Hazards and Earth System Sciences*) 20th-anniversary survey, where 350 natural hazard community members responded to two questions: (Q1) "What are the top three scientific challenges you believe are currently facing our understanding of natural hazards?" and (Q2): "What three broad step changes should or could be done by the natural hazard community to address natural hazards in achieving the Sustainable Development Goals?" We have analysed the data quantitatively and qualitatively. According to the 350 respondents, the most significant challenges (Q1) are the following [with in brackets, % of 350 respondents who identified a given theme]: (i) shortcomings in the knowledge of risk and risk components [64 %], (ii) deficiencies of hazard and risk reduction approaches [37 %], (iii) influence of global change, especially climate change [35 %], (iv) integration of social factors [18%], (v) inadequate translation of science to policy and practice [17 %], and (vi) lack of interdisciplinary approaches [6 %]. In order for the natural hazard community to support the implementation of the Sustainable Development Goals (Q2), respondents called for: (i) enhanced stakeholder engagement, communication, and knowledge transfer [39 %], (ii) increased management and reduction of disaster risks [34 %], (iii) enhanced interdisciplinary research and its translation to policy and practice [29 %], (iv) a better understanding of natural hazards [23 %], (v) better data, enhanced access to data and data sharing [9 %], and (vi) increased attention to developing countries [6 %]. We note that while the most common knowledge gaps are felt to be around components of knowledge about risk drivers, the step changes that the community felt were necessary related more to issues of wider stakeholder engagement, increased risk management and interdisciplinary working.

==GRAPHICAL ABSTRACT HERE==

## 1 Introduction

In their interaction with people, economies, and the built and natural environment, natural hazards result in small- and large-scale disasters (Sidle et al., 2004; Luebken and Mauch, 2011). Understanding these hazards and devising creative ways to manage them and their impacts has always attracted interest from scientists, policy makers, and practitioners (Peek and Mileti,

2002). However, disaster risks are still mounting due to exacerbating climate change, increasing populations, accelerated urbanisation, and land-use change (Coronese et al., 2019; Glasser, 2020; WMO, 2021), amongst other factors and processes. Therefore, it is of interest to assess gaps in understanding and management of natural hazards and the resulting risks. Given the importance of the United Nations Sustainable Development Goals (SDGs) (UNISDR, 2015a) as the world's blueprint for sustainable futures, it is also noteworthy to explore the relationship between natural hazards sciences and their implementation (Gill and Smith, 2021). This includes considering the changes needed to overcome knowledge gaps to contribute to the implementation of the SDGs. This paper will focus on these natural hazard issues by providing some perspectives of the broader European Geosciences Union (EGU) natural hazard community. We do this through a questionnaire launched as part of the 20th anniversary of Natural Hazards and Earth System Sciences (NHESS). This paper summarises the responses of 350 natural hazard scientists, government workers and practitioners whom we asked via a questionnaire: (i) the top scientific challenges in our understanding of natural hazards, (ii) suggested step changes by the natural hazard community to address natural hazards in achieving the SDGs.

Over the past five decades, there have been large leaps and advancements in our scientific understanding of natural hazards and their management. There is an extensive breadth of disciplines and fields involved in research on natural hazards (e.g., engineering, physical and social sciences, humanities); thus, covering all advancements would be beyond this paper's scope. Following, we give eight exemplars of mechanisms by which these advancements have been facilitated, by no means exhaustive, based on discussions between the authors of this paper:

- The development and enhancement of physical, statistical, and numerical modelling of different natural hazards enable us to better understand the processes behind the generation and propagation of natural hazards and the characterisation of different natural hazards (OECD, 2012; Hirabayashi et al., 2013; Strauch et al., 2019).
- Increasing computational power allows for better, faster, and more complex modelling based on the higher spatial and temporal resolution of data (Ujjwal et al., 2019).
- Access to new and innovative data sources and data analysis techniques, such as (i) remote sensing and Geographical Information Systems (GIS) (Gillespie et al., 2007; Poursanidis and Chrysoulakis, 2017), (ii) data and data generated through citizen science (Cochran et al., 2009; Paul et al., 2018), and (iii) qualitative GIS representations of resilience (Taylor et al., 2020).
- Systematic collection of parameters related to different natural hazards through the development of measurement technology (Angeli et al., 2000; Boiten, 2003; Herrmann et al., 2011).
- Development of and advancement in early warning systems for natural hazards, including all of its four components (as defined by UNISDR, 2006) of (i) risk knowledge, (ii) monitoring and warning service, (iii) dissemination and communication, (iv) response capability (Šakić Trogrlić et al., 2022).
- Increasing focus on multi-hazard characterisation and management of multi-risks (Gill and Malamud, 2014; Pescaroli and Alexander, 2018; Ward et al., 2022; Kreibich et al., 2022a), increasing communications between different

traditionally 'single hazard' groups (e.g., the UK Natural Hazards Partnership, Hemingway and Gunawan, 2018) and synergies and asynergies between management options for different natural hazards (de Ruiter et al., 2021).

- Increased understanding of the critical interaction of natural hazards with vulnerability shaped by social, economic, political, and cultural processes (O'Keefe et al., 1976; Cutter and Finch, 2008; Chmutina and von Meding, 2019), and with exposure (Lerner–Lam, 2007; Iglesias et al., 2021).
- The development of global policies to deal with disaster risks, the latest being the Sendai Framework for Disaster Risk Reduction 2015–2030 (UNISDR, 2015b).

These advancements have been made possible due to scientific research across disciplines. This paper uses a broad classification of science, including natural sciences, engineering and technology, medical and health sciences, agricultural sciences, social sciences, and humanities (OECD, 2007). We also view research related to natural hazards and the reduction and management of related risks as inherently interdisciplinary (Peek and Guikema, 2021; Johnston and van de Lindt, 2022). Science is seen as key to policy-making for disaster risk reduction (Aitsi–Selmi et al., 2016a) and an imperative to meet the need for Disaster Risk Reduction (DRR) (Shi et al., 2020). Science for DRR is crucial for understanding/characterising risk components (i.e., hazard, exposure, vulnerability), communicating disaster risk, and informing risk management and reduction options (JRC, 2017).

Despite increasing understanding of the importance of science in reducing disaster risks and undeniable scientific advancement, literature recognises many outstanding challenges in understanding hazards and risks and managing/reducing them (Albris et al., 2020; Shi et al., 2020). For instance, in their recent paper, Cui et al. (2021) identify six frontier scientific issues:

- Hazard-formation mechanisms due to multi-spherical interaction and identification of potential disaster risk.
- Disaster mechanisms corresponding to the coupling of endogenic and exogenic processes.
- Evolution of natural hazards and compound disaster chain generation.
- Temporal-spatial evolution of disaster risk.
- Theory of integrated disaster risk management.
- Development of a resilient social model of harmony between man and nature.

Cui et al. (2021) also point out critical technological gaps, including monitoring and forecasting natural hazards, dynamic risk assessment, and comprehensive governance. Wartman et al. (2020) bring attention to significant challenges in modelling different natural hazards, decision-making under uncertainty, building community resilience, and mitigation of disaster risks. Similarly, Aitsi–Selmi et al. (2016a) agree on the gaps in hazard and risk modelling, but also identify a need for more efficient risk information sharing and capacity building across different stakeholders. Additional challenges were identified in the NHESS special issue "*Perspectives on challenges and step changes for addressing natural hazards*" (Kreibich et al., 2022b); for instance, from the perspectives of reinsurance (Rädler, 2022) and specific countries (Germany: Thiebs et al. 2022; Italy: Simonelli et al. 2022; Switzerland: Wabbes and Bezzola, 2022).

Overcoming these challenges and gaps can result in risk reduction and resilience building of people and nations, thus supporting the implementation of the SDGs. The SDGs comprise 17 global goals and were jointly agreed upon by 193 countries in September 2015 (Aitsi-Selmi et al., 2016b). These goals serve as a blueprint for both developed and developing countries on how to create sustainable futures by 2030. According to the reflection paper by UNISDR (2015a), there are 25 targets related to DRR in 10 out of 17 global goals. Consequently, a need for an enhanced understanding of how the natural hazard community can support the implementation of the SDGs is evident.

This paper is organised as follows: **Section 2** presents our methodological approach, including survey design and implementation, and thematic analysis of received responses. **Section 3** presents our results, including the overview of survey respondents and an analysis of the response to the two questions. **Section 4** discusses some of the findings, while in **Section 5**, we draw the main conclusions.

## 2 Methods

This section describes our methodology for our survey design and implementation (**Section 2.1**) and then our data analysis (**Section 2.2**).

### 2.1 Survey design and implementation

The four executive editors of the journal *Natural Hazards and Earth System Sciences* (NHESS, https://nhess.copernicus.org/), and in the context of the 20th anniversary of the journal, designed the text for a questionnaire (see Appendix A) which was then placed in an online format using Google forms with the following three parts:

- Part 1. General introduction and how participant responses will be used
- Part 2. Two questions, with space for open-ended answers:
  a) Question 1 (Q1). What are the top three scientific challenges you believe are currently facing our understanding of natural hazards?
  b) Question 2 (Q2): What three broad step changes should or could be done by the natural hazard community to address natural hazards in achieving the Sustainable Development Goals?
- Part 3. Identification preferences of the participants (i.e., their institution name and country, position, name).

A brief message advertising the NHESS questionnaire (along with its URL link) was then shared through (i) an e-mail to the mailing list of those affiliated with the EGU natural hazard division (1550 e-mails), (ii) an e-mail to the EGU NHESS author contact list (3085 e-mails, some which are no longer current), (iii) the EGU division Facebook page (994 people following), (iv) the EGU Natural Hazard division Twitter feed (1900 followers), and (v) some personal Twitter feeds. There is a clear overlap in the different methods being used. Still, we estimate that the advertising reached 2000 to 4000 people, primarily natural scientists (including students), who self-identified as part of the natural hazard community. The EGU contact lists based on contact authors of the NHESS publications are primarily European (71 %), but with a substantial minority from North

America (13 %), Asia (12 %), and then fewer from other global regions (4 %) such as Africa and South America. The questionnaire was left open for two months, and in total, there were 350 responses.

## 2.2 Data analysis

Respondents' answers to Questions 1 and 2 were open-ended and qualitative, resulting in a rich data set. Upon the initial screening of the answers and understanding of the diversity of community views, we analysed the data set using thematic analysis. Thematic analysis is a commonly used method in analysing qualitative data, based on identifying, analysing, and reporting themes within data sets (Braun and Clarke, 2006). In our case, thematic analysis was beneficial for summarising large data sets and identifying key features (King, 2004). We had 350 diverse answers from respondents in different disciplines and geographical regions, presenting discipline and geography-focused views. Therefore, thematic analysis was an approach allowing us to come up with meaningful themes describing the data.

For the analysis, we used QSR NVivo version 1.5, a qualitative data software (NVivo, 2021). We chose to use this software for data analysis given the large amount of data collected, as it allowed for an accessible overview of data, coding process, and manipulation of sub-themes and themes. Paulus and Lester (2020) note that using software to analyse qualitative data allows for different data management options and leads to more efficient and effective analysis. Using software also allows the researcher to analyse a greater quantity of data, faster and more complexly (Robins and Eisen, 2017).

In NVivo, we did the following:

- The first author assigned codes to pieces of text, including words, phrases, sentences and paragraphs (Miles and Huberman, 1994).
- We then merged codes into sub-themes and sub-themes into themes.
- We conducted the above separately for Questions 1 and 2.

Overall, in our thematic analysis, we employed the following six stages of thematic analysis as outlined by Nowell et al. (2017):

1. **Familiarising ourselves with the data**: Upon uploading the data set to NVivo, we first explored the data through critical reading and understanding the most used phrases.
2. **Generating initial codes**: Initially, the first author created 58 codes for Question 1 and 38 codes for Question 2, which were then discussed and revised by all authors.
3. **Searching for themes**: Upon revising the codes and grouping them into sub-themes, followed by merging themes into sub-themes, we created 11 themes for Question 1 and 9 themes for Question 2.
4. **Reviewing themes**: We then reviewed the themes in online meetings and re-grouped them into six themes for both questions.
5. **Defining and naming themes**: Similar to the above, we refined the names of the themes and defined what is included under each of the themes.
6. **Producing the report**: The results were written up and presented in **Section 3**.

Upon finalising the thematic analysis in NVivo, we continued the analysis in Excel. One aspect we were interested in was regional perspectives. We first went through all information on institutions given by respondents, and from each of these identified their region. Themes in NVivo were then expressed according to the percentage of total responses from the region.

## 3 Results

**Section 2** elaborated on how we collected the responses and our general approach to the analysis. In this section, we begin by presenting respondents' demographics (**Section 3.1**), followed by an analysis of respondents' identification of top scientific challenges facing the natural hazards community (**Section 3.2**), and finally, the broad step changes needed in the natural hazards community to address the SDGs (**Section 3.3**).

### 3.1 Overview of survey respondents

In total, we received 350 responses. In terms of regional representation (Table 1a), the majority of respondents were from European countries ($n = 201$ [57 %] out of 350), with Southern and Western Europe both showing the strongest response rates, followed by Asia ($n = 38$ [11 %]) and North America ($n = 25$ [7 %]). In contrast, there were fewer responses from the rest of the regions (South and Central America, Caribbean, Oceania and Africa), which we grouped together ($n = 27$ [7.7 %]. In addition, 59 respondents did not indicate their regional profile. The institutional profile of the participants is provided in Table 1b with the majority of respondents representing academia ($n = 198$ [57 %]), national research centres, institutes and labs ($n = 48$ [14 %]), and governmental organizations ($n = 29$ [8 %]).

TABLE 1 HERE

We did not explicitly ask the respondents what hazards are their primary point of interest (either scientifically or practically). However, to gain some understanding of hazards of interest for respondents, we identified, in their answer to Question 1, instances when they explicitly mentioned specific hazards, as presented in Figure 1. In total, 123 out of 350 respondents explicitly mentioned a hazard. We used the following hazards grouping:

- *geophysical hazards* (earthquakes, landslides, tsunamis, volcanoes)
- *hydrological hazards* (floods, droughts, also includes water availability)
- *atmospheric* (cold and heat waves, hurricanes, sandstorms, cyclones, rain, hail, storms, tornadoes, typhoons)
- *marine hazards* (sea level rise, sea surges, waves, coastal flooding, saltwater intrusion, coastal retreat)
- *biophysical* (wildfires)
- *environmental hazards* (desertification, deforestation, environmental pollution, soil erosion, karstification, loss of biodiversity)

FIGURE 1 HERE

### 3.2 Analysis of Question 1: Top three scientific challenges currently facing our understanding of natural hazards?

Respondents' answers on the scientific challenges facing our understanding of natural hazards were grouped under six broad and distinct themes. Figure 2 provides an overview of the themes, indicating the percentage of respondents (% out of $n = 350$) whose answers were classified under this theme, and providing some example quotes.

==FIGURE 2 HERE==

We now briefly describe each of the six themes for Question 1, including the main characteristics of the answers for each of the themes we chose, using thematic analysis in NVivo as described in Section 2.2.

**Q1. Theme 1A Shortcomings in knowledge of risk and risk components**

The largest proportion of participants' answers (64 % of 350 respondents) identified shortcomings in the knowledge of risk and risk components (i.e., hazard, exposure, vulnerability) as the most significant scientific challenge facing our understanding of natural hazards. The following were some of the main characteristics identified in respondent answers in terms of knowledge shortcomings:

- There are major gaps in existing knowledge on the characterisation of different hazards (e.g., non-stationarity of natural hazards, choices of return periods, spatial-temporal patterns of extreme events, non-linearity of processes).

- There is limited knowledge of vulnerability and exposure, especially their dynamics. For instance, Respondent 96 (position, location not indicated) noted "*understanding and modelling changes and dynamics in vulnerability*'' as a significant challenge.

- There is a lack of sophistication and accuracy of modelling approaches (e.g., current status of physics-based models, available methods for model validation, and characterisation of uncertainty in modelling). Modelling interactions of different risk components remains a challenge; as described by Respondent 135 (Professor, Belgium), "*modelling co-evolution of hazard and socio-economic dynamic*s". Furthermore, respondents' answers pointed to the inadequacy of data informing the modelling process (e.g., inadequate spatial and temporal resolution of the existing datasets).

- There is a prominent gap in our understanding of risk and its components was identified as the lack of understanding of multi-hazards and multi-risk, especially the interrelationships between different hazards. Respondents identified cascading, compound, and triggering interrelationships. For example, Respondent 14 (Associate Professor, USA) pointed out that "*cascading and overlapping events present challenges for traditional hazard planning*''.

- Finally, there is a paucity in the characterisation of risk to impact pathways was noted, including different types of impact (e.g., impact on social systems).

**Q1. Theme 1B Deficiencies of hazard and risk management approaches**

Answers from 37 % of the 350 respondents indicated that they see deficiencies of hazard and risk management approaches as one of the key challenges faced by the natural hazards community. Primarily, this theme was dominated by describing deficiencies in current forecasting of natural hazards (e.g., validation of forecasts, forecasting models, forecast lead times) and

especially forecasting of earthquakes (although other hazards such as floods, droughts, meteorological phenomena, and landslides were also indicated).

This theme also contained answers on the deficiencies in knowledge on various approaches to risk management. For instance, Respondent 96 (position, location not indicated) identified as one of the key challenges "*development and assessment of ecosystem-based approaches to disaster risk reduction and climate change adaptation*'', while some others felt that "*operational and customised early warning system*s'' are missing (Respondent 188, Director, Cuba).

**Q1. Theme 1C Influence of global change, especially climate change**

Not surprisingly, 35 % of the 350 respondents pointed to general gaps in understanding how various processes of global change influence natural hazards and vice versa. This predominantly focused on climate change, but also included land-use change, urbanisation, and population growth (with the latter three influencing exposure to natural hazards).

For climate change, respondents pointed out gaps in our understanding of climate change impacts on the frequency and magnitude of natural hazards, how climate change will influence their spatio-temporal behaviour, and various uncertainties augmented by climate change (e.g., in modelling hazard scenarios).

**Q1. Theme 1D Integration of social factors**

Answers from 18 % of the 350 respondents indicated that social-aspects of hazard and risk management are not fully integrated into natural hazards risk analysis. For instance, our current ability to integrate human-related factors into modelling processes, as emphasised by Respondent 18 (Associate Professor, USA) "*Our understanding of and ability to model feedbacks in coupled-human natural systems*.'' Further comments indicated that the natural hazards community recognised that current hazard and risk analysis lacks integration of risk perception, local and indigenous knowledge systems, and the overall consideration of inequality in understanding the risks of natural hazards.

**Q1. Theme 1E Inadequate translation of science to policy and practice**

Interestingly, 17 % of the 350 respondents also recognised that the challenge might not be in our understanding of natural hazards, but rather in the inadequacy in the way science is communicated and scientific knowledge transferred to different stakeholders, including gaps in science-policy interface (SPI) and translation of research findings to practical approaches. For instance, many responses classified under this theme were concerned with how to communicate scientific information to a vast array of stakeholders, with communication challenges also including how we communicate forecasts and warnings.

Furthermore, respondents drew attention to several aspects related to the interface of science, policy and practice. These included the scientific community not having policy experience; science not "infiltrating'' policy and decision-making spaces and informing policy design; research findings not informing and being translated to practical implementation.

**Q1. Theme 1F Lack of interdisciplinary approaches**

The final theme comprises 6 % of the 350 respondents, who pointed out that although natural hazards and DRR are per se interdisciplinary, there is a lack of interdisciplinary research, approaches, and professional capacity.

**3.3 Analysis of Question 2: Three broad step changes that should or could be done by the natural hazard community to address natural hazards in achieving the Sustainable Development Goals**

In the previous section, we presented challenges identified by the natural hazards community in our understanding of natural hazards and related risks. This section describes what broad step changes identified by the 350 respondents that the natural hazards community could do to address natural hazards in achieving the Sustainable Development Goals (SDGs).

Before addressing the suggested changes identified, we were interested in understanding the percentage of respondents' answers which explicitly referred to a specific SDG or could be inferred to refer to a specific SDG. Out of 350 respondents, 188 (54 %) mentioned one or more SDGs, either explicitly by name/number or we interpreted it from the answer they gave to Question 2. Figure 3 shows the results (out of 188) for the 17 SDGs identified by participants. The majority of responses (40 % of 188) referred to the natural hazard community contribution to the implementation of the SDG 13: Climate Action, followed by SDG 11: Sustainable Cities and Communities (25 %), SDG 17: Partnership for the Goals (24 %), and Quality Education (24 %). Five other SDGs (SDG 15, SDG 1, SDG 9, SDG 10, and SDG6) had 10–13 % of participants referring to them, while 8 SDGs were referred to by 2–9 % of respondents.

==FIGURE 3 HERE==

Respondents' answers on the broad changes needed in the natural hazards community to contribute to SDG implementation were grouped under six broad and distinct themes. Figure 4 provides an overview of the themes, indicating the percentage of respondents (% out of $n = 350$) whose answers were classified under each theme, as well as providing example quotes.

==FIGURE 4 HERE==

We now briefly describe each of the six themes for Question 2, including the main characteristics of the answers for each of the themes we chose, using thematic analysis in NVivo as described in Section 2.2.

**Q2. Theme 2A Enhanced stakeholder engagement, communication, and knowledge transfer**

The largest theme (39 % of 350 respondents) refers to answers identifying a need for enhanced, better and active participation of different stakeholder groups in research activities and decision-making (e.g., involvement of communities at risk in the design of DRR strategies, collaboration with decision-makers in policy making). For example, Respondent 175 (Associate Professor, Italy) thought there is a need for ''*getting* [scientists] *more involved in the decision-making process*.''

This theme also emphasises a need to communicate science and scientific findings to the general public and other actors. This involved research in risk communication and facilitating knowledge transfer horizontally (e.g., between scientific disciplines) and vertically (from scientists to other stakeholders, and local communities ''upwards'').

Finally, respondents felt an urgency to increase risk awareness in communities at risk and general education of the public on the topic of risks of natural hazards. For example, Respondent 141 (Associate Professor, Brazil) pointed out a need for "*active participation of communities in understanding the phenomena associated with disasters.*" Further practical advice given by respondents was to introduce natural hazards education in schools, do public outreach and education, and design educational programmes focused on natural hazards.

**Q2. Theme 2B Increased management and reduction of disaster risks**

A total of 34 % of 350 respondents provided answers focusing on a need to improve the way risks of natural hazards are managed and reduced. The largest sub-theme was an urgency for enhanced monitoring and forecasting of natural hazards. For example:

- More accurate forecasts with a shorter lead time,
- Establishment of global monitoring and early warning systems,
- Calls for hazard-specific forecasting (e.g., seasonal drought prediction, earthquake forecasting).

Under this theme, respondents also called for improvements in various approaches to managing hazards and risks beyond mere forecasting and monitoring. These include, for instance, an enhanced understanding of nature-based solutions, "*more consideration of natural hazards in spatial and urban planning"* (Respondent 266, Research Scientist, Australia), and "*combination of structural and non-structural measures* [...] *to achieve the SDGs*." (Respondent 238, Research Fellow, USA).

**Q2. Theme 2C Enhanced interdisciplinary research and its translation to policy and practice**

Based on answers from 29 % of 350 participants, this theme summarises calls for more research that will facilitate filling in of identified knowledge gaps and calls for increased research funding. Special attention was given to calls for interdisciplinarity and multidisciplinarity (e.g., removing the siloes of scientific disciplines and encouraging more collaboration between disciplines). Some examples include:

- Interdisciplinary research addressing multiple SDGs simultaneously,
- Interdisciplinary postgraduate studies.

Also, respondents identified an urgent need for research to be better translated to policy and practice (e.g., bridging between the scientific community, practitioners, and decision-makers). For instance, Respondent 113 (Professor, Italy) pointed to a need to "*enhance the transfer of scientific results into concrete actions*." Further examples of solutions proposed by respondents included:

- Creation of basic services based on robust research findings,
- Creation of decision support environments informed by the latest research,
- Research informing government policies and practice,
- Need for applied research.

**Q2. Theme2 D Better understanding of natural hazards**

A total of 23 % of 350 respondents identified a need to enhance our understanding of various aspects of natural hazards (e.g., spatio-temporal characterisation of natural hazards, a common definition of hazard magnitudes and frequencies). This theme also includes respondents who proposed the following solutions by the natural hazard community in achieving the SDGs:

- Enhancement of hazard analysis methodologies (e.g., understanding changes in natural hazards), modelling of natural hazards and associated uncertainties (e.g., increasing computation skills, black swan phenomenon, use of new technologies in natural hazard modelling),
- Characterisation of multi-hazards (e.g., assessments of multi-risks, hazard interrelationships),

- Inclusion of social factors in understanding hazards (e.g., understanding resilience, inequality consideration in risk understanding).

**Q2. Theme 2E Better data, enhanced access to data and data sharing**

There were 9 % of 350 respondents who saw a way forward in achieving the SDGs by better data for characterisation and modelling of natural hazards, including a need to utilise emerging areas of data research such as big data. The theme also reflects calls for open data and data sharing between different scientific communities and geographies, as reflected in the answer by Respondent 250 (Senior scientist, Switzerland): "*Open all environmental monitoring and observation data for everybody and any purpose as soon as possible after acquisition.*"

**Q2. Theme 2F Increased attention to developing countries**.

Finally, 6 % of 350 respondents drew attention to the fact that there needs to be an enhanced focus on understanding natural hazards in the context of developing countries. This includes more research on natural hazards in these contexts and more collaboration with partners from the Global South.

**3.4 Some reflections on regional differences in answers to Questions 1 and 2**

This section examines regional differences in the response percentages to Questions 1 and 2, noting that observed differences are marginal due to relatively low responses in each regional category. Using the eight regional categories presented in Figure 1, Table 2a presents the regional responses to Q1 on the top challenges facing our understanding of natural hazards. Table 2b presents the regional responses to Q2 on the step changes recommended by the natural hazards community to address natural hazards in achieving the SDGs.

==TABLE 2 HERE==

From Table 2a, we observe that for Question 1, on the top challenges facing our understanding of natural hazards, most regions generally had higher percentages of responses in A, followed by B and C, with similar values within each theme for most regions. The exception was Asia, which had a much lower response to Theme A compared to other regions, with Theme C being a similar value. It is noteworthy that these Themes A to C are primarily concerned with an absence of physical scientific knowledge and were identified by most participants as being key challenges. This is compared to the relatively few responses classified under Themes D to F, that noted the importance of social factors, policy issues or interdisciplinary approaches in their responses to this question.

Most responses to Question 2 (Table 2b) were coded in themes A and B across the regions. Relatively few respondents suggested the need for Theme F, increased attention to developing countries, but those that did tend to be either from the regions of South and Central America, Caribbean, Oceania, Africa or Western Europe. North American respondents tended not to note issues with data (Theme E). Interdisciplinary working (Theme C) was noted by slightly fewer people in Eastern and Northern Europe. We also observe that although a relatively low percentage (average 7 %) noted lack of interdisciplinary approaches as a challenge in Q1 (Table 2a, Theme F), a much larger percentage (average 29 %) identified the need for interdisciplinary research to address natural hazards in the context of the SDGs in Q2 (Table 2b, Theme C).

## 4 Discussion

Through thematic analysis of 350 responses to two questions, we identified some views of the natural hazards community on the scientific challenges facing our understanding of natural hazards (Section 3.2) and broad changes needed to support the implementation of the SDGs (Section 3.3).

We found that challenges are many and cut across different scientific disciplines and fields of expertise. The challenges identified in our work align well with challenges identified in the literature (e.g., Aitsi-Selmi et al., 2016b; Wartman et al., 2020; Zuccaro et al, 2020; Cui et al., 2021). However, our analysis was based on responses from a wider natural hazard community ($n = 350$) compared to previous studies, thus representing views of diverse types of stakeholders that make up the natural hazard community, including researchers, academics, and representatives of the governmental organisations. For instance, Cui et al. (2021) based their findings on a systematic analysis of literature, similar to Shi et al. (2020). Other studies report findings based on the analysis of stakeholders' responses, but more focused. For example, Freddi et al. (2021) focus on the views of the earthquake DRR community, while Wartman et al. (2020) present the views of the reconnaissance community.

We see that many challenges identified by our respondents are timely and in line with big scientific questions that the natural hazards community is currently facing. For instance, Theme D (Integration of Social Factors) also includes calls for further understanding of the interaction between human societies and natural hazards, a theme covered in growing fields like socio-hydrology (e.g., Di Baldassarre et al., 2013; Mazzoleni et al., 2021). Similarly, there are growing calls for increased understanding of multi-hazards, their interrelationships, and their impact, a topic of increased focus within the scientific community (e.g., Gill and Malamud, 2016; de Ruiter et al., 2020; Ward et al., 2022; De Angeli et al., 2022; Kreibich et al., 2022a).

Although our Question 1 focused on scientific challenges facing understanding of natural hazards, answers provided by respondents go beyond natural hazards and towards more holistic risk-thinking and existing gaps (e.g., in terms of risk reduction approaches, inadequate translation of science to policy and thinking, lack of inclusion of social factors). Thus, the major theme for Question 1 is "Shortcomings in knowledge of risk and risk components", where participants identified existing knowledge gaps not only of hazard processes but also in terms of our understanding of vulnerability, exposure, and impact. This might suggest a need for greater engagement between the natural hazards community and the social sciences.

Moreover, although challenges identified in Question 1 are often of a more "technical" and "scientific" nature (e.g., hazard forecasting, modelling approaches), answers to Question 2 and major themes emerging had a slightly different focus. For instance, according to our respondents, the major step forward was an identified need for enhanced stakeholder engagement, communication, and knowledge transfer. This was closely followed by a shift towards the actual reduction and management of disaster risks (e.g., through nature-based solutions, impact-based forecasting), interdisciplinary research and its translation to policy and practice. This change of focus from "technical" and "scientific" to "managerial" and "governance-oriented" shows that respondents felt that, for the natural hazards community to enhance the implementation of the SDGs, is not necessarily about the next scientific break-through, but about overcoming common obstacles in the delivery of DRR (e.g., top-

down approaches, lack of engagement with different stakeholders, and policy and practice not informed by science) which are beyond just scientific realm.

Our results also show no cause-effect relationships between themes identified for Q1 and Q2. For instance, although 64 % of respondents' answers were classified under Q1 (challenges) Theme 1A "Shortcomings in our knowledge of risk and risk components", only 23 % of answers were classified under Q2 (step changes needed) Theme 2D "Better understanding of natural hazards". This can be explained by the fact that Q2 focused on the possible changes needed in natural hazard research and practice to contribute to the implementation of the SDGs. Moreover, it can be explained by the point raised in the previous paragraph of the results showing that natural hazard contributions to the implementation of the SDGs are not only about scientific break-throughs, but more about the larger changes in policy and practice of DRR.

Our findings on the way forward indicate that the natural hazards community can contribute to all 17 SDGs. Most writings on the connection between DRR and SDGs identify fewer links. For instance, UNISDR (2015a) mentions DRR-related targets in 10 of 17 SDGs (missing SDG5, SDG7, SDG8, SDG10, SDG11, SDG16, SDG17), while Izumi et al. (2020) identify linkages with 14 SDGs (missing SDG7, SDG12, and SDG16). It is apparent from our findings that natural hazard communities see their role in each of the SDGs, both in terms of research (e.g., characterising links between climate change and natural hazards), but also in terms of delivering concrete solutions to challenges tackled by SDGs (e.g., nature-based solutions for climate change mitigation and adaptation). This is consistent with the suggestions of articles within Gill and Smith (2021). One of the reasons our analysis identifies links with all 17 SDGs could be the open nature of our questions and a wide range of responses from respondents with a mix of scientific disciplines and institutional backgrounds. Furthermore, it could be due to an increasing understanding of a need for coherence between guiding global policies, including the SDGs, Sendai Framework, Paris Agreement, and the New Urban Agenda.

It is especially interesting to see a clear recognition of the importance of interdisciplinary research on natural hazards in achieving the SDGs. It is widely accepted that challenges related to sustainable development cannot be solved under the realm of any specific discipline. For instance, this has previously been discussed in research on health (Herzig Van Wees et al., 2019), geosciences (Gill and Bullough, 2017; Gill and Smith, 2021) and land use (Johansen et al., 2020). Our results suggest the same for the natural hazards community. Gill and Smith (2021) contain articles that give examples of how the geosciences more broadly relate to the SDGs, including work on natural hazards (e.g., Gill et al., 2021; Smith and Bricker, 2021).

But as identified by natural hazard community members, it is crucial how to translate this research to policy and practice. Challenges to implementing the SDGs are many; for instance, addressing different SDGs in parallel and limited financial resources in developing countries (Akenroye et al., 2018). Leal Filho et al. (2021) identify a lack of policies, proper governance, financial resources, training programmes and trained personnel as key challenges in this area.

Taken together, responses to both questions suggest that while there are many important technical and scientific challenges around the forecasting and understanding of natural hazards, there are also felt to remain gaps in the integration of natural hazards knowledge with other disciplines, such as the social sciences and their integration beyond the academy. This points to

a desire for more robust and effective transdisciplinary approaches to ensure that disaster science is "useful, usable and used" (Boaz and Hayden, 2002; Aitsi-Selmi et al., 2016a).

Our findings indicate that even with a physical-science oriented set of participants, the themes of challenges in Q1 that go beyond the realm of typical physical science were deemed important (e.g., Integration of social factors, Inadequate translation of science to policy and practice, Lack of interdisciplinary approaches). These can also be attributed to the rising prominence of interdisciplinarity in natural hazards and disaster studies (Peek and Guikema, 2021). This rise has occurred as researchers

have become increasingly aware that these issues are social ones as much as they are natural, and require collaboration between natural scientists, social scientists, medicine and the humanities, amongst others. Some of this concern arises from experience, in which scientists have struggled with issues of communication or recognised the role played by vulnerability in the outworking of particular crises and have extended their interests beyond single discipline science. Examples include Barclay et al. (2008) which builds on experience from multiple volcanic crises; and Bretton et al. (2015) which tackles issues arising from the L'Aquila earthquake.

Over the past 40 years, the UN disasters office has transitioned from the UN International Decade for Natural Disaster Reduction to the Sendai Framework for Disaster Risk Reduction. Comparing these policy approaches demonstrates the shift from physical-science heavy approaches towards interdisciplinarity, with the recognition that "disasters are not natural". The UNDRR documents such as the Sendai Framework are explicit about this, and feed into the policy of national governments, creating a parallel push for interdisciplinary working from the policy environment. Many projects now seek to integrate multiple disciplines, with varying degrees of success, and exploring the best ways to do so remains a topic of active research

(Morss et al., 2021).

This transition from hazard-based approaches to risk-based interdisciplinary approaches is also reflected in the NHESS publications since 2003. We examined the change over time in five-year increments of keyword phrases (ranging from one to four words) in all NHESS papers, from 2003 to 2022 (through to 31 March 2022). Keywords used were from Clarivate (2022) *Web of Science* 'Keywords Plus' which reflect the titles of the references cited by the authors of each NHESS paper. We exclude 2001 and 2002, as the number of keywords for both years were not recorded for a high percentage of papers and

together were fewer than the first three months of 2022. Out of a total for the 20 years of 3965 unique keyword phrases (3110 papers that had keyword+ information), 1-word phrases were used 42 % of the time, 2 words 44 %, and 3 words 14 %. The period 2003–2007 averaged 4.2 keywords per paper, increasing to 7.4 keywords per paper in 2018–2022. We manually combined plural/singular keywords into one form, and combined similar words into one word or phrase (e.g., 'rain', 'rainfall', 'precipitation', 'heavy rain(s)', all became 'precipitation'; 'impact(s)', 'loss(es)', 'damage(s)' all became 'impact'; 'hurricane(s)', 'cyclone(s)', 'typhoon(s)' all became 'tropical storm'.

We present our results in four word-clouds in Fig. 5, where the higher the percentage of a word appearing relative to all words for that five-year period, the larger the word appears in the word-cloud. We observe that from 2003–2007 to 2008–2012 to 2013–2017 to 2018–2022 the following changes (all percentages given are with respect to the keyword phrase occurrence weighting for that time period).

- 'Climate Change' steadily becomes more prominent over the four time periods: 2003–2007 (**0.2 %**), 2008–2012 (**0.6 %**), 2013–2017 (**1.1 %**), 2018–2022 (**1.6 %**).

- 'Risk' steadily becomes more prominent over the four time periods: 2003–2007 (**0.5 %**), 2008–2012 (**1.3 %**), 2013–2017 (**1.8 %**), 2018–2022 (**2.5 %**).

- 'Impact' becomes more prominent in 2013–2017 and 2018–2022: 2003–2007 (**1.4 %**), 2008–2012 (**1.2 %**), 2013–2017 (**2.6 %**), 2018–2022 (**3.0 %**).

- 'Vulnerability' becomes more prominent in 2013–2017 and 2018–2022: 2003–2007 (**0.6 %**), 2008–2012 (**0.6 %**), 2013–2017 (**1.1 %**), 2018–2022 (**1.2 %**).

The first bullet points reflects the increased focus on climate change in papers, and the last three bullet points the change from hazard- to risk-based approaches.

==FIGURE 5 HERE==

The limitations of this research are primarily connected to the respondent profiles and analysis method. As presented in Section 3.1, respondents were primarily from Europe, with a significantly lower representation from other regions. Therefore, it is essential to acknowledge that the views represented could be biased towards the views of European participants and that an

468 increased number of respondents from other regions would quite possibly modify the themes. Furthermore, 16.9 % of respondents did not indicate their region, and 18.0 % did not indicate their institutions. Finally, NHESS is a journal with a predominant focus on physical science, meaning that the views of, for instance, social scientists, were not captured extensively in our analysis. The results would likely have been very different, for example, had this survey gone to a broader community of disaster risk researchers, which might include additional social, medical and agricultural scientists as well as a broader range of practitioners.

We have used thematic analysis, which is ultimately a subjective analysis method, albeit widely used in qualitative data analysis. It depends on the researchers' positionality (e.g., epistemological and ontological stance). The group discussion of the themes and the dataset makes it intersubjective, and we acknowledge that a different set of authors would likely have come up with slightly different themes. However, group discussion of the themes also served as a way to reduce the subjectivity and come up with agreed themes that would represent the diversity of data. Answers from participants were very diverse, and in the majority of cases, the answers were concise. Therefore, it was challenging to develop really "strict" themes with

"impermeable" boundaries. Although the identified themes were quite broad, they still conveyed the main priorities identified by respondents and depict the richness and diversity of inputs.

**5 Conclusions**

In this paper, we have presented the natural hazards community views on the most significant scientific challenges regarding natural hazards and the broad changes needed to contribute to the implementation of the SDGs. To examine the natural hazards community views, we have used thematic analysis of the responses from 350 natural hazard scientists, government workers,

and practitioners. We have classified the responses for each of the two questions into six themes. According to the 350 respondents, the most significant challenges are the following: (i) shortcomings in the knowledge of risk and risk components, (ii) deficiencies of hazard and risk reduction approaches, (iii) influence of global change, especially climate change, (iv) integration of social factors, (v) inadequate translation of science to policy and practice, and (vi) lack of interdisciplinary approaches. For the natural hazard community to support the implementation of SDGs, respondents called for: (i) enhanced stakeholder engagement, communication, and knowledge transfer, (ii) increased management and reduction of disaster risks, (iii) enhanced interdisciplinary research and its translation to policy and practice, (iv) a better understanding of natural hazards, (v) better data, enhanced access to data and data sharing, and (vi) increased attention to developing countries.

Based on our analysis, we suggest the following actionable steps that could be (and in several countries already are existing good practice) taken in supporting the implementation of the SDGs:

- Continued progress towards better data, enhanced access to data, and data sharing in the context of natural hazards, risk and DRR. Increased funding for mapping the quality, flow (and barriers) of DRR related quantitative and qualitative data, particularly in Global South contexts, would help to underpin better a lot of the research currently being done to understand natural hazards and risk drivers.

- Continued fundamental and applied research in the various aspects of understanding natural hazards and, more widely, risk drivers.

- Increased funding of interdisciplinary research focused on contributions of natural hazard research to the implementation of the SDGs. Our findings could help to inform the focus of these additional funding calls. There are still considerable challenges in interdisciplinary practice that require dedicated research into integrating disciplines, particularly across the social and physical sciences, but also in some cognate disciplines (e.g., natural hazard scientists with statisticians). This might suggest the need for additional resources to learn from past projects in this area.

- Enhanced and continued collaboration between natural hazard scientists, decision-makers, and wider actors (e.g., civil society) in charge of implementing the SDGs. This could be done through, for instance, the establishment of science-technology–policy networks that could be useful in supporting policy implementation (Sakic Trogrlic et al., 2017). It would also mean higher involvement of scientists and social scientists in the existing decision-making spaces.

The results of the thematic analysis point to scientific challenges within the natural hazard community that remain numerous and cut across scientific disciplines and topics of interest. Identified challenges in this paper could provide valuable and useful insights for further research and inform tailored research funding. As for the natural hazard community supporting the SDG implementation, our analysis shows that scientific advancements are just a part of the solution. At the same time, primary changes are needed in the way risks are governed (e.g., who gets a chance to participate and how is knowledge transferred) and how science informs policy and practice.

**Appendices:**

**Appendix A Online Survey Form**

FIGURE A1 HERE

## Data availability

We provide as supplementary material an Excel file with the raw data obtained from the questionnaire (see Appendix A) for the 350 natural hazard community member respondents, with respondent names and institutions removed. The data are classified according to two questions and answers are split into three columns for each of the two questions. The Excel file also contains information on each respondent's job title, along with our interpretation (based on their institution name, not given in this Excel sheet) of their institution type, country and region.

## Supplement link.

The link to the supplement will be included by Copernicus.

## Author contributions

RST, BDM, and AD designed the methodology. RST conducted the formal analysis with inputs and regular discussions with BDM and AD. RST took the lead on preparing the original draft with BDM and AD contributing, and then all authors jointly reviewed and edited subsequent drafts.

## Disclaimer

Publisher's note: Copernicus Publications remains neutral with regard to jurisdictional claims in published maps and institutional affiliations.

## Competing interests

Authors BDM and ARD are members of the editorial board of NHESS. The peer-review process was guided by an independent editor, and the authors also have no other competing interests to declare."

## Special issue statement

This article is part of the special issue "Perspectives on challenges and step changes for addressing natural hazards". It is not associated with a conference.

## Acknowledgements

We gratefully acknowledge all 350 respondents in the natural hazards community for taking the time to fill in the questionnaire and provide data for this study. The authors also thank the following NHESS executive editors who, together with author BDM, helped with questionnaire development and dissemination: Heidi Kreibich (GFZ), Uwe Ulbrich (Freie Universität Berlin) and Paolo Tarolli (University of Padua).

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

Table 1: Overview of the (a) regional profile and (b) institutional profile of the 350 natural hazard community member respondents to the NHESS questionnaire on challenges to our understanding of natural hazard challenges and step changes the natural hazard community might make in the context of the Sustainable Development Goals. The region given in (a) is as defined by United Nations (2021).

 Figure 1: Major hazard groups identified by natural hazard community members to the NHESS survey question Q1 "What are the top three scientific challenges you believe are currently facing our understanding of natural hazards", together with example quotes. The percentages given are with respect to the total number of respondents whose answers mentioned a specific hazard (n = 123), not all respondents (n = 350).

Figure 2: Six themes identified through thematic analysis of responses of 350 natural hazard community members to the NHESS survey question Q1 "What are the top three scientific challenges you believe are currently facing our understanding of natural hazards?" along with example quotes. Percentages indicate the percentage of all respondents (n = 350) whose answers were classified under a given theme.

Figure 3: Percentage of n = 188 natural hazard community members referring to specific United Nations Sustainable Development Goals (SDG) in response to NHESS survey question Q2 on broad step changes that the natural hazards community could make to achieve the SDGs. The percentages have been determined out of the total number of respondents that specified an SDG (n = 188), not all respondents (n = 350). SDG icons from United Nations (2019).

 Figure 4: Six themes identified through thematic analysis of responses of 350 natural hazard community members to the NHESS survey question Q2 "What three broad step changes should or could be done by the natural hazard community to address natural hazards in achieving the Sustainable Development Goals?" along with example quotes. Percentages indicate the percentage of all respondents (n = 350) whose answers were classified under a given theme.

Table 2. Regional difference in responses of 350 natural hazard community members to the two NHESS survey questions: (a) Q1 top challenges facing our understanding of natural hazards and (b) step changes to address natural hazards in achieving the Sustainable Development Goals (SDGs). Numbers given are expressed as the percentage of the total number of respondents in a region whose answers were classified under a specific theme.

Figure 5. Word clouds of NHESS (Natural Hazards and Earth System Sciences) article keyword phrases for four five-year periods: (a) 2003–2007 (597 unique keyword phrases, 292 articles with keywords), 2008–2012 (1,682 unique keywords, 1018 articles), 2013–2017 (1,970 unique keywords, 989 articles), 2018–2022 (1,757 unique keywords, 811 articles). Keywords as given by Clarivate (2022) Web of Science Keywords Plus, a reflection of the reference titles cited by each NHESS paper. For each time period, the top 200 unique keyword phrases are shown. Similar words were combined manually and only one form of plural/singular was maintained. Word clouds created in WordClouds.com (2022).

Figure A1. Online survey form for the NHESS 20th anniversary Questionnaire in Google Forms to which 350 natural hazard community members responded.

## Invited perspectives: Views of 350 natural hazard community members on key challenges in natural hazards research and the Sustainable Development Goals

| QUESTION ASKED | THEMES IDENTIFIED THROUGH THEMATIC ANALYSIS (expressed in percentages of the 350 respondents identifying given theme) | | | | | |
|---|---|---|---|---|---|---|
| Q1. "What are the top three scientific challenges you believe are currently facing our understanding of natural hazards?" | 1A: Shortcomings in knowledge of risk and risk components | 1B: Deficiencies of hazard and risk reduction approaches | 1C: Influence of global change, especially climate change | 1D: Integration of social factors | 1E: Inadequate translation of science to policy and practice | 1F: Lack of interdisciplinary approaches |
| | 64% | 37% | 35% | 18% | 17% | 6% |
| Q2. "What three broad step changes should or could be done by the natural hazard community to address natural hazards in achieving the Sustainable Development Goals?" | 2A: Enhanced stakeholder engagement, communication, and knowledge transfer | 2B: Increased management and reduction of disaster risks | 2C: Enhanced interdisciplinary research and its translation to policy and practice | 2D: Better understanding of natural hazards | 2E: Better data, enhanced access to data and data sharing | 2F: Increased attention to developing countries |
| | 39% | 34% | 29% | 23% | 9% | 6% |

Natural Hazards and Earth System Sciences — Open Access — EGU

Robert Šakić Trogrlić, Amy Donovan, Bruce D Malamud

Trogrlić *et al.* (2022) https://doi.org/10.5194/nhess-2022-55

798

**Graphical abstract (DOI will need updating) and key figure for online.**

**Table 1: Overview of the (a) regional profile and (b) institutional profile of the 350 natural hazard community member respondents to the NHESS questionnaire on challenges to our understanding of natural hazard challenges and step changes the natural hazard community might make in the context of the Sustainable Development Goals. The region given in (a) is as defined by United Nations (2021).**

| (a) Region | Respondent # (% of total) | | Countries [# respondents] |
|---|---|---|---|
| Southern Europe | 90 (26%) | ▬ | Italy [42], Spain [23], Greece [13], Portugal [6], Croatia [4], Slovenia [1] |
| Western Europe | 77 (22%) | ▬ | Germany [29], France [20], Switzerland [9], The Netherlands [7], Belgium [6], Austria [6] |
| Northern Europe | 22 (6%) | ▪ | United Kingdom [16], Norway [5], Sweden [1] |
| Eastern Europe | 12 (3%) | ▪ | Russia [4], Romania [3], Bulgaria [2], Poland [2], Czech Republic [1] |
| Asia | 38 (11%) | ▪ | China [11], Japan [8], India [7], Turkey [3], Israel [2], Iraq [2], Nepal [2], Georgia [1], Kazakhstan [1], Thailand [1] |
| South & Central America, Caribbean, Oceania, Africa | 27 (8%) | ▪ | Australia [5], Brazil [5], Colombia [4], Mexico [3], Chile [2], Ecuador [2], New Zealand [2], Egypt [1], Ethiopia [1], Costa Rica [1], Cuba [1] |
| North America | 25 (7%) | ▪ | United States [21], Canada [4] |
| Region not indicated | 59 (17%) | ▪ | |
| Total | 350 (100%) | | |

| (b) Institution | Respondent # (% of total) | |
|---|---|---|
| University | 198 (57%) | ▬ |
| National Research Centres, Institutes and Labs | 48 (14%) | ▪ |
| Governmental Organization | 29 (8.3%) | ▏ |
| International Organization | 6 (1.7%) | │ |
| International Research Centre | 5 (1.4%) | │ |
| Non-Governmental Organization | 1 (0.3%) | |
| Institution not indicated | 63 (18%) | ▪ |
| Total | 350 (100%) | |

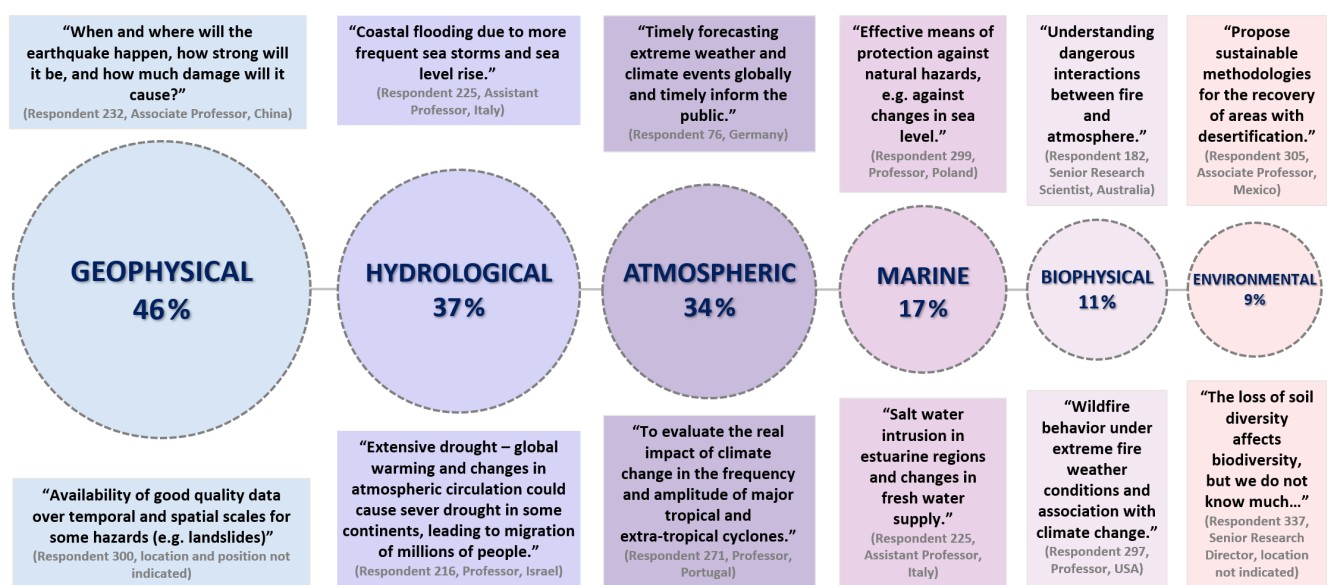

**Figure 1: Major hazard groups identified by natural hazard community members to the NHESS survey question Q1 "What are the top three scientific challenges you believe are currently facing our understanding of natural hazards", together with example quotes. The percentages given are with respect to the total number of respondents whose answers mentioned a specific hazard (*n* = 123), not all respondents (*n* = 350).**

810

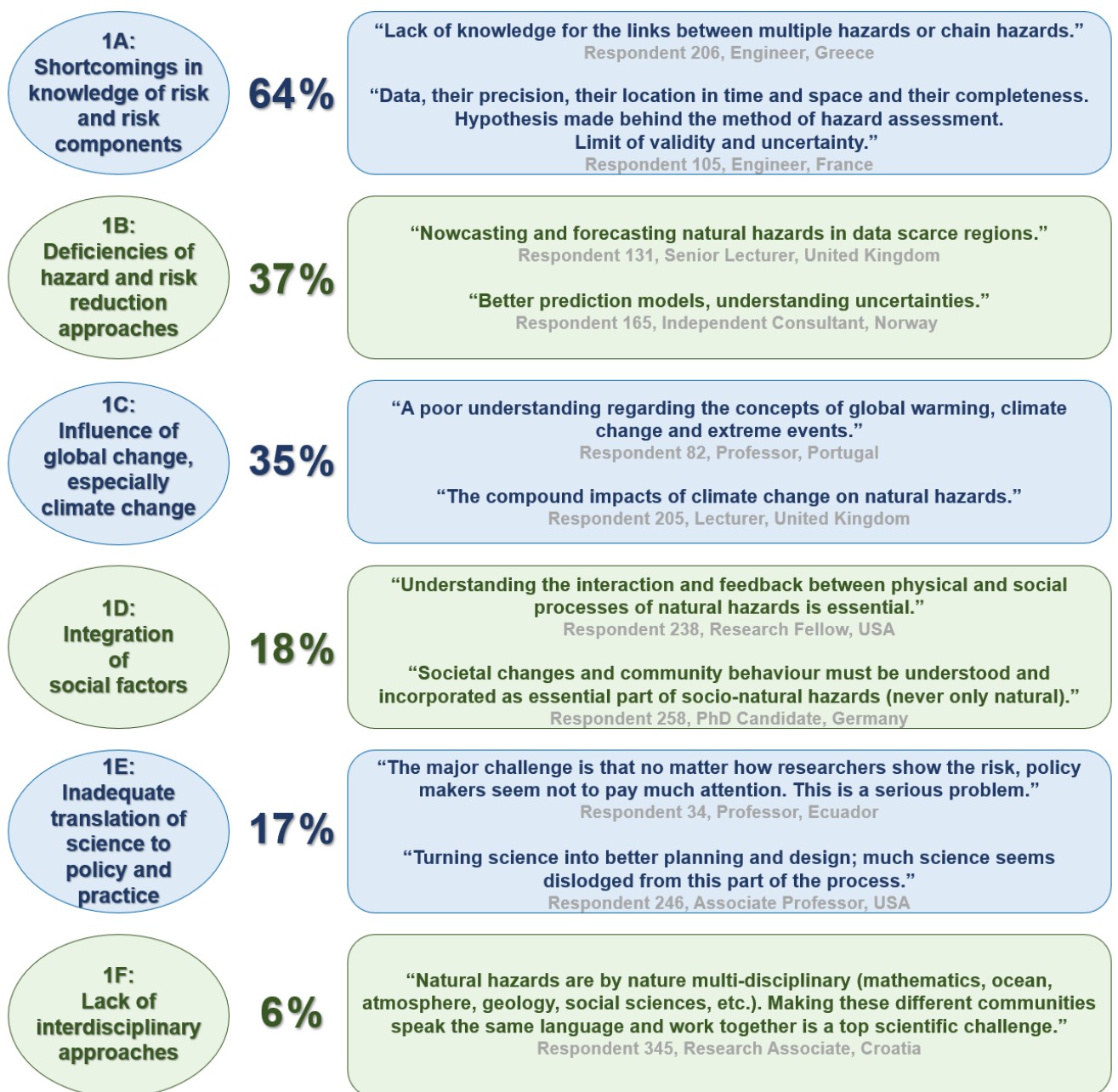

Figure 2: Six themes identified through thematic analysis of responses of 350 natural hazard community members to the NHESS survey question Q1 "What are the top three scientific challenges you believe are currently facing our understanding of natural hazards?" along with example quotes. Percentages indicate the percentage of all respondents (*n* = 350) whose answers were classified under a given theme.

816

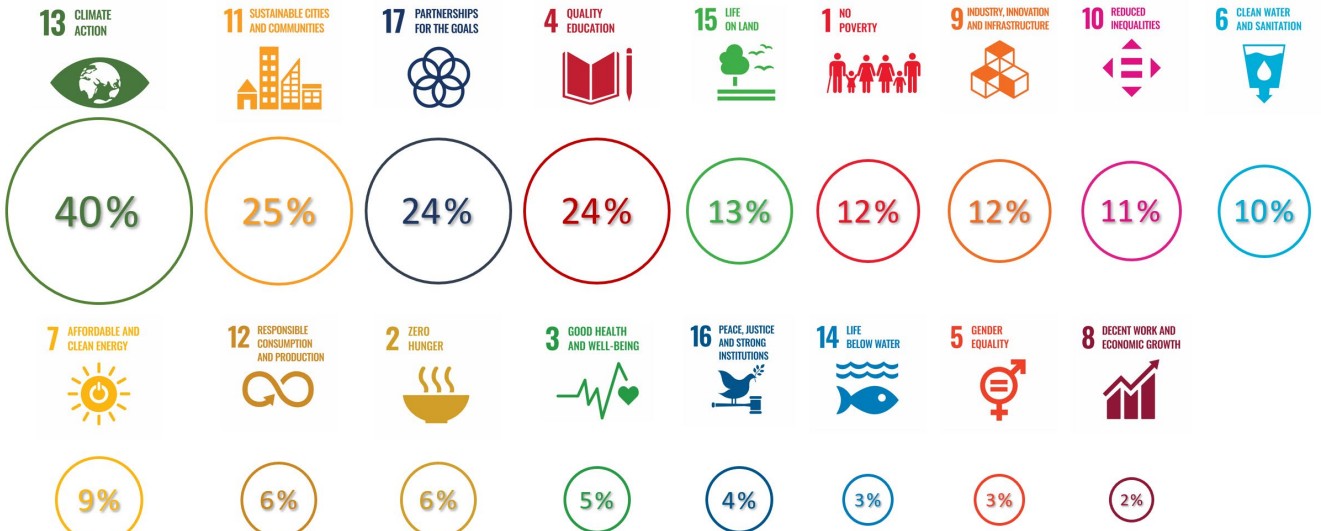

**Figure 3: Percentage of *n* = 188 natural hazard community members referring to specific United Nations Sustainable Development Goals (SDG) in response to NHESS survey question Q2 on broad step changes that the natural hazards community could make to achieve the SDGs. The percentages have been determined out of the total number of respondents that specified an SDG (*n* = 188), not all respondents (*n* = 350). SDG icons from United Nations (2019).**

822

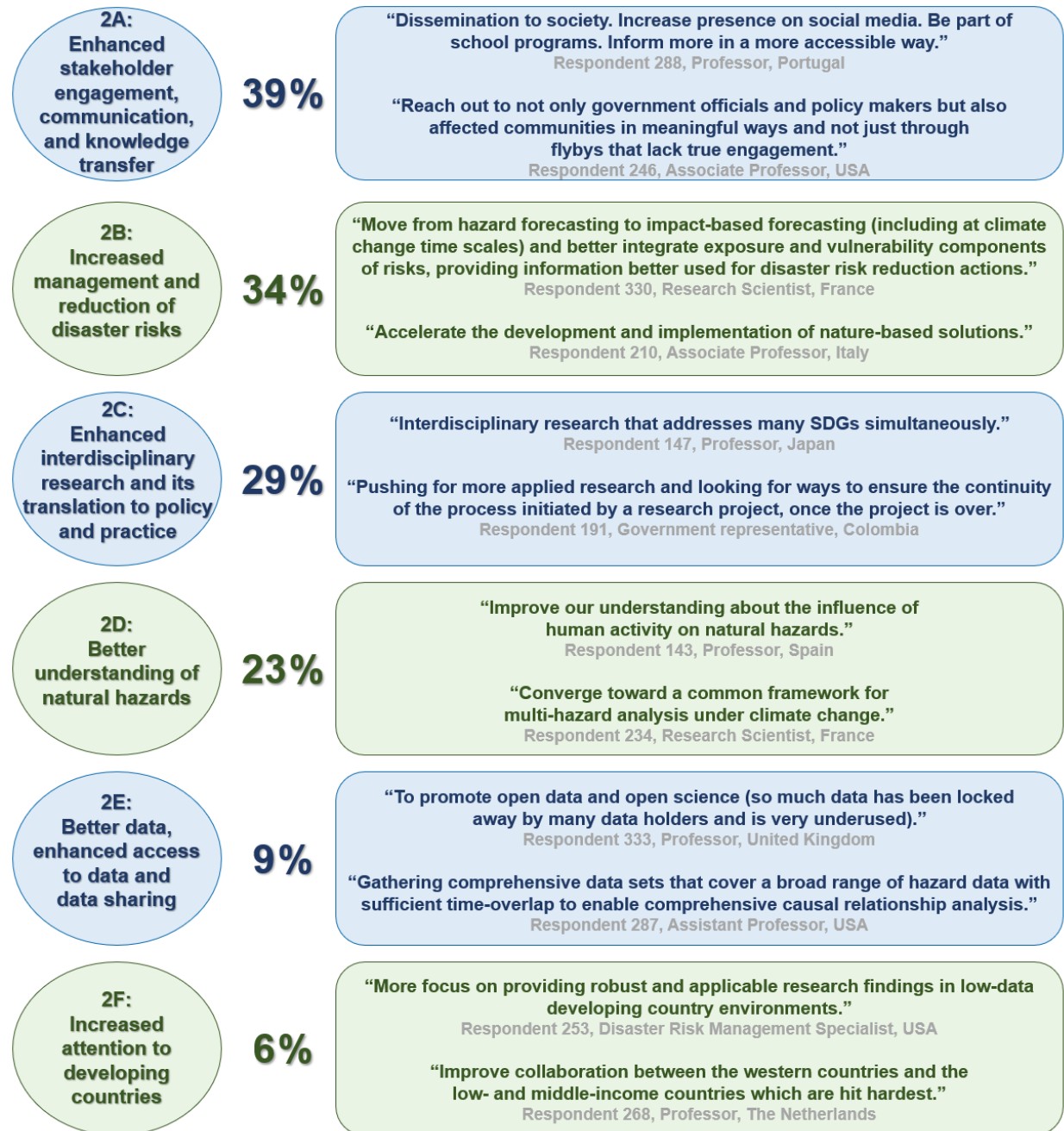

**Figure 4: Six themes identified through thematic analysis of responses of 350 natural hazard community members to the NHESS survey question Q2 "What three broad step changes should or could be done by the natural hazard community to address natural hazards in achieving the Sustainable Development Goals?" along with example quotes. Percentages indicate the percentage of all respondents (*n* = 350) whose answers were classified under a given theme.**

828

**Table 2. Regional difference in responses of 350 natural hazard community members to the two NHESS survey questions: (a) Q1 top challenges facing our understanding of natural hazards and (b) step changes to address natural hazards in achieving the Sustainable Development Goals (SDGs). Numbers given are expressed as the percentage of the total number of respondents in a region whose answers were classified under a specific theme.**

| (a) Q1. Top challenges facing our understanding of natural hazards | 1A: Shortcomings in knowledge of risk and risk components | 1B: Deficiencies of hazard and risk reduction approaches | 1C: Influence of global change, especially climate change | 1D: Integration of social factors | 1E: Inadequate translation of science to policy and practice | 1F: Lack of interdisciplinary approaches |
|---|---|---|---|---|---|---|
| Southern Europe (n = 90) | 58 % | 44 % | 40 % | 20 % | 19 % | 7 % |
| Western Europe (n = 77) | 69 % | 35 % | 32 % | 17 % | 10 % | 3 % |
| Northern Europe (n = 22) | 73 % | 45 % | 50 % | 18 % | 27 % | 5 % |
| Eastern Europe (n = 12) | 83 % | 42 % | 25 % | 33 % | 17 % | 8 % |
| Asia (n = 38) | 37 % | 26 % | 37 % | 16 % | 8 % | 11 % |
| South & Central America, Caribbean, Oceania, Africa (n = 27) | 81 % | 44 % | 26 % | 7 % | 11 % | 11 % |
| North America (n = 25) | 72 % | 40 % | 40 % | 12 % | 28 % | 4 % |
| Unassigned (n = 59) | 64 % | 27 % | 27 % | 20 % | 24 % | 7 % |
| Average across eight regions | 67 % | 38 % | 35 % | 18 % | 18 % | 7 % |
| Standard deviation across eight regions | 15 % | 8 % | 9 % | 8 % | 8 % | 3 % |
| Percentage of total respondents (n = 350) | 64 % | 37 % | 35 % | 18 % | 17 % | 6 % |

| (b) Q2. Step changes to address natural hazards in achieving the SDGs | 2A: Enhanced stakeholder engagement, communication, and knowledge transfer | 2B: Increased management and reduction of disaster risks | 2C: Enhanced interdisciplinary research and its translation to policy and practice | 2D: Better understanding of natural hazards | 2E: Better data, enhanced access to data and data sharing | 2F: Increased attention to developing countries |
|---|---|---|---|---|---|---|
| Southern Europe (n = 90) | 44 % | 29 % | 28 % | 20 % | 9 % | 2 % |
| Western Europe (n = 77) | 34 % | 40 % | 31 % | 27 % | 16 % | 10 % |
| Northern Europe (n = 22) | 41 % | 36 % | 18 % | 27 % | 18 % | 5 % |
| Eastern Europe (n = 12) | 42 % | 33 % | 17 % | 33 % | 8 % | 0 % |
| Asia (n = 38) | 46 % | 46 % | 36 % | 25 % | 11 % | 0 % |
| South & Central America, Caribbean, Oceania, Africa (n = 27) | 37 % | 41 % | 33 % | 22 % | 0 % | 19 % |
| North America (n = 25) | 52 % | 40 % | 36 % | 28 % | 4 % | 4 % |
| Unassigned (n = 59) | 32 % | 25 % | 32 % | 19 % | 7 % | 5 % |
| Average across eight regions | 41 % | 36 % | 29 % | 25 % | 9 % | 6 % |
| Standard deviation across eight regions | 7 % | 7 % | 8 % | 5 % | 6 % | 6 % |
| Percentage of total respondents (n = 350) | 39 % | 34 % | 29 % | 23 % | 9 % | 6 % |

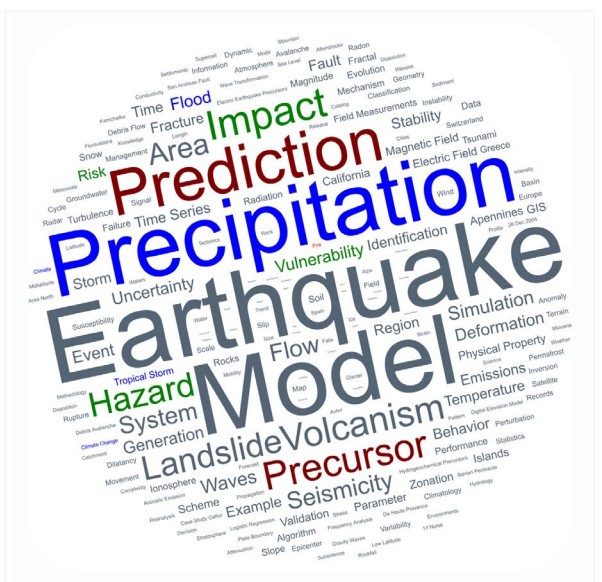

**(a) 2003−2007**

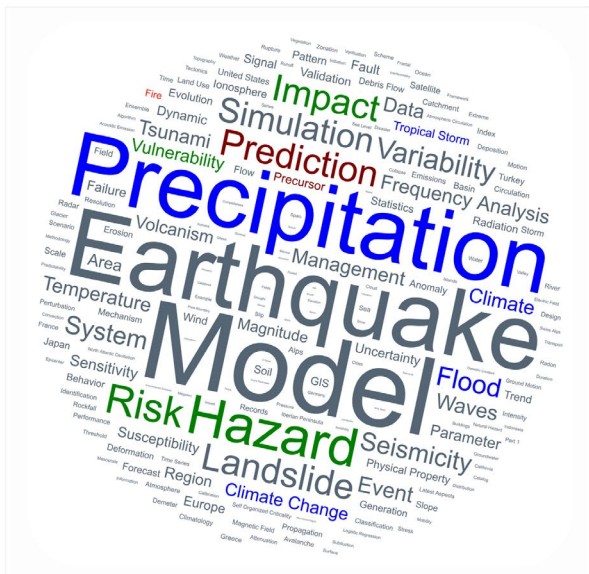

**(b) 2008−2012**

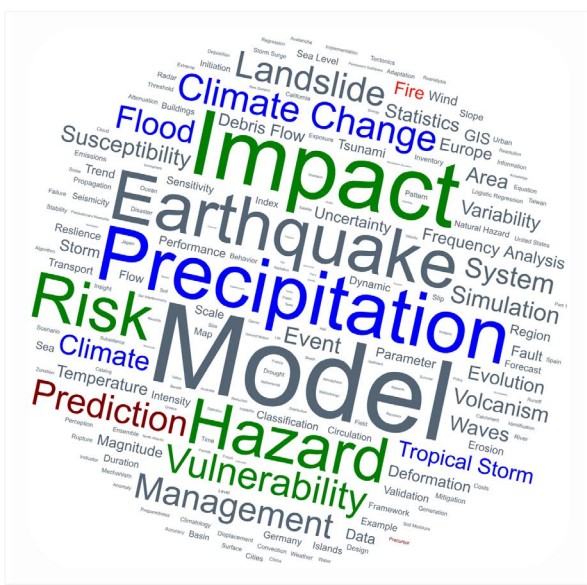

**(c) 2013−2017**

**(d) 2018−2022**

840

**Figure 5. Word clouds of NHESS (*Natural Hazards and Earth System Sciences*) article keyword phrases for four five-year periods: (a) 2003–2007 (597 unique keyword phrases, 292 articles with keywords), 2008–2012 (1,682 unique keywords, 1018 articles), 2013–2017 (1,970 unique keywords, 989 articles), 2018–2022 (1,757 unique keywords, 811 articles). Keywords as given by Clarivate (2022) *Web of Science* Keywords Plus, a reflection of the reference titles cited by each NHESS paper. For each time period, the top 200 unique keyword phrases are shown. Similar words were combined manually and only one form of plural/singular was maintained. Word clouds created in WordClouds.com (2022).**

846

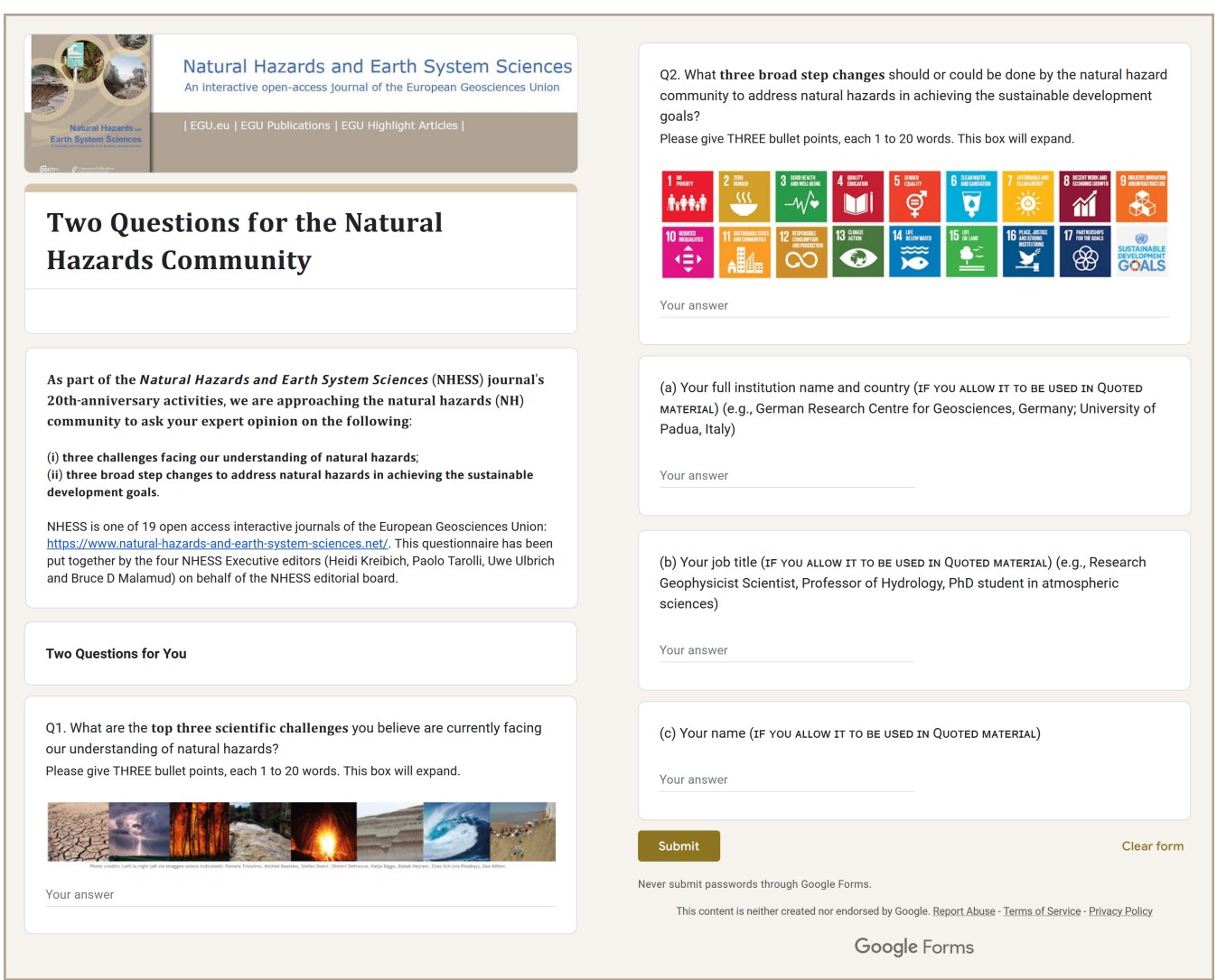

**Figure A1. Online survey form for the NHESS 20<sup>th</sup> anniversary Questionnaire in Google Forms to which 350 natural hazard community members responded.**