# Peer review of "Invited perspective: Views of 350 natural hazard community members on key challenges in natural hazards research and the Sustainable Development Goals"

_Natural Hazards and Earth System Sciences, 2022_

## Author Comment (AC2)

**Reply to reviewer comments on NHESS-2022-55**

We thank the editor and the two reviewers (R1, R2) for their very helpful comments on our manuscript **"Invited perspective: A community perspective on natural hazards key challenges and the Sustainable Development Goals"** by Robert Šakić Trogrlić, Amy Donovan and Bruce D Malamud, submitted on 15 February 2022 to *Natural Hazards and Earth System Sciences* (Manuscript nhess-2022-55). Below, we provide detailed replies to each of the reviewer's comments. Lines numbers below refer to the original version submitted. We believe that the resultant changes have improved the manuscript, and again thank the reviewers.

**Reviewer 1 (R1)**

**(R1-00):** The manuscript (ID: nhess-2022-55) mainly analyzed two questions about natural hazards via the journal (NHESS) Online Survey Form from 350 natural hazard scientists, government workers, and practitioners. Through the analysis of the questionnaire, the authors identified the most significant scientific challenges to the understanding of natural hazards and step changes to achieving sustainable development goals. This work is very interesting and important since it provides perspectives of the natural hazard community on the SDGs after 7 years into its launch. The paper is well written with good logic and well-organized structures. I would recommend publishing with minor revisions. My main suggestions for improvement are as follows:

**(Reply to R1-00):** We thank R1 for these comments and below respond to each of the reviewer main suggestions.

**(R1-01):** From Table 3, (Q1 Theme 1A) and (Q2 Theme 2D): these two themes look like cause and action, but the results seem inconsistent. 64% of the respondent identified "shortcoming in the knowledge of risk and risk components (Q1 Theme 1A)" as the main challenger but only 23% put "better understanding of natural hazards(Q2 Theme 2D)" as the steps change to achieve. If lack of knowledge is a problem, then steps need to be taken to generate that knowledge. The author may want to elaborate on these. A similar case can be found between Q1 Theme 1E and Q2 Theme 2A.

**(Reply to R1-01):** We thank the reviewer for this comment. In addition to the paragraph we wrote in L420-L430 (original manuscript) that touched upon the change of focus from more ''technical'' themes identified in Q1 to broader themes identified in Q2 (e.g., stakeholder engagement, science-policy-practice interface, and wider governance-related aspects), we have now added the following as our 6th paragraph of the discussion:

"Our results also show no cause-effect relationships between themes identified for Q1 and Q2. For instance, although 64% of respondents' answers were classified under Q1 (challenges) Theme

1A "Shortcomings in our knowledge of risk and risk components", only 23% of answers were classified under Q2 (step changes needed) Theme 2D "Better understanding of natural hazards". This can be explained by the fact that Q2 focused on the possible changes needed in natural hazard research and practice to contribute to the implementation of the SDGs. Moreover, it can be explained by the point raised in the previous paragraph of the results showing that natural hazard contributions to the implementation of the SDGs are not only about scientific break-throughs, but more about the larger changes in policy and practice of DRR."

**(R1-02):** The article has made many interesting findings. Maybe authors could take it further to discuss the issues behind these findings in order to better guide the research. For example, in Line 408, the article indicated the interaction between human societies and natural hazards and emphasized its increasing significance in scientific research. Readers would be more interested to know the reason behind this trend and in your opinion, how to integrate these two components in scientific research.

**(Reply to R1-02):** We agree that the issues behind these findings is in and of itself very interesting, although believe to go into depth this would be another paper. To partially address the reviewer's comment, we have added in the following.

[Added two paragraphs towards the Discussion end, also wrote this in response to R2-06]

"Our findings indicate that even with a physical-science oriented set of participants, the themes of challenges in Q1 that go beyond the realm of typical physical science were deemed important (e.g., Integration of social factors, Inadequate translation of science to policy and practice, Lack of interdisciplinary approaches). These can also be attributed to the rising prominence of interdisciplinarity in natural hazards and disaster studies (Peek and Guikema, 2021). This rise has occurred as researchers have become increasingly aware that these issues are social ones as much as they are natural, and require collaboration between natural scientists, social scientists, medicine and the humanities, amongst others. Some of this concern arises from experience, in which scientists have struggled with issues of communication or recognised the role played by vulnerability in the outworking of particular crises and have extended their interests beyond single discipline science. Examples include Barclay et al. (2008) which builds on experience from multiple volcanic crises; and Bretton et al. (2015) which tackles issues arising from the L'Aquila earthquake.

Over the past 50 years, the UN disasters office has transitioned from the UN International Decade for Natural Disaster Reduction to the Sendai Framework for Disaster Risk Reduction. Comparing these policy approaches demonstrates the shift from physical-science heavy approaches towards interdisciplinarity, with the recognition that "disasters are not natural". The UNDRR documents such as the Sendai Framework are explicit about this, and feed into the policy of national governments, creating a parallel push for interdisciplinary working from the policy environment.

Many projects now seek to integrate multiple disciplines, with varying degrees of success, and exploring the best ways to do so remains a topic of active research (Morss et al., 2021).''

**(R1-03):** Line 431, a comparison of the numbers of identified links between natural hazards community and SDGs was presented. The author may further discuss which SDGs are the increased linkage from UNDRR (2015) and Izumi (2020) and why. This could provide insight on the progress since the laughing of SDGs.

**(Reply to R1-03):** We have now added in the following:

- Which SDGs were not covered in Izumi et al. (2020) and UNDRR (2015).
- A reflection on the increased number of linkages (in the 7$^{th}$ paragraph of the discussion): 'One of the reasons our analysis identifies links with all 17 SDGs could be the open nature of our questions and a wide range of responses from respondents with a mix of scientific disciplines and institutional backgrounds. Furthermore, it could be due to an increasing understanding of a need for coherence between guiding global policies, including the SDGs, Sendai Framework, Paris Agreement, and the New Urban Agenda. ''

**(R1-04):** In the conclusion, this paper listed the challenges and required effort to support the implementation of SDGs. The author may improve them by elaborating the six calls into more actionable steps instead of providing general directions.

**(Reply to R1-04):** We have now added the following as the second to last paragraph of the conclusions.

Based on our analyses, we suggest the following actionable steps that could be (and in several countries already are existing good practice) taken in supporting the implementation of the SDGs:

- Continued progress towards better data, enhanced access to data, and data sharing in the context of natural hazards, risk and DRR. Increased funding for mapping the quality, flow (and barriers) of DRR related quantitative and qualitative data, particularly in Global South contexts, would help to underpin better a lot of the research currently being done to understand natural hazards and risk drivers.
- Continued fundamental and applied research in the various aspects of understanding natural hazards and, more widely, risk drivers.
- Increased funding of interdisciplinary research focused on contributions of natural hazard research to the implementation of the SDGs. Our findings could help to inform the focus of these additional funding calls. There are still considerable challenges in interdisciplinary practice that require dedicated research into integrating disciplines, particularly across the social and physical sciences, but also in some cognate disciplines (e.g., natural hazard scientists with statisticians). This might suggest the need for additional resources to learn from past projects in this area.

- Enhanced and continued collaboration between natural hazard scientists, decision-makers, and wider actors (e.g., civil society) in charge of implementing the SDGs. This could be done by, for instance, in the establishment of science-technology–policy networks which could be useful in supporting policy implementation (Sakic Trogrlic et al., 2017). It would also mean higher involvement of scientists and social scientists in the existing decision-making spaces.

**(R1-05):** Line 472 and 473: Typo on the symbols '' . Please modify it.

**(Reply to R1-05):** We made this change.

**(R1-06):** Table 3: the 3rd row of "REGION," the region's name is not shown.

**(Reply to R1-06):** We have replaced the Table 3 with a corrected version.

**Reviewer 2 (R2)**

**(R2-00):** The manuscript shows perspectives from the wider natural hazards community on key challenges within their scientific field. The study builds on a questionnaire survey conducted online with the journal's wider community. The results are consistent with the relevant literature, but add little new knowledge. However, they provide an additional comprehensive picture of the natural hazard community. The manuscript is overall well written and comprehensible. I recommend a minor revision prior to publication. In detail:

**(Reply to R2-00):** Thank you for these comments.

**(R2-01):** Line 14: The reader would have a better introduction to the manuscript if the authors already added the key findings (like in line 480-486) briefly in the abstract.

**(Reply to R2-01):** We have now added this in the abstract.

**(R2-02):** Line 33: Who do the authors mean by "our"? The authors of the manuscript? The community? Or the current state of science? Please specify.

**(Reply to R2-02):** This is clarified now by adding the word ''scientific''.

**(R2-03):** Line 34: What is the background to this list? Is it based on a discussion between the authors or a comprehensive analysis of the literature? Even if it is stated as not complete, adding some more items might be necessary. One thing to mention is the issue of warnings and forecasts (e.g. the importance of social media), which is of deeper relevance in the later course of the manuscript and the discussion of the results. Overall, a deeper engagement with existing literature on key challenges would be useful.

**(Reply to R2-03):** In the second paragraph of the introduction, we have expanded on the background of the list by adding "Over the past five decades, there have been large leaps and advancements in our scientific understanding of natural hazards and their management. There is an extensive breadth of disciplines and fields involved in research on natural hazards (e.g., engineering, physical and social sciences, humanities); thus, covering all advancements would be beyond this paper's scope. Following, we give eight exemplars of mechanisms by which these advancements have been facilitated, by no means exhaustive, based on discussions between the authors of this paper:"

Directly after this we have added another two bullet points (in addition to some amendments to the others):

- Development of and advancement in early warning systems for natural hazards, including all of its four components (as defined by UNISDR, 2006) of (i) risk knowledge, (ii) monitoring and warning service, (iii) dissemination and communication, (iv) response capability (Šakić Trogrlić et al., 2022).
- Increasing focus on multi-hazard characterisation and management of multi-risks (Gill and Malamud, 2014; Pescaroli and Alexander, 2018; Ward et al., 2022; Kreibich et al., 2022), increasing communications between different traditionally 'single hazard' groups (e.g., the UK Natural Hazards Partnership, Hemingway and Gunawan, 2018) and synergies and asynergies between management options for different natural hazards (de Ruiter et al., 2021).

**(R2-04):** Line 56: The abbreviation DRR is used here for the first time without further explanation.

**(Reply to R2-04):** We have now added the full abbreviation.

**(R2-05):** Line 139 (Point 4): Was the analysis carried out equally by all authors? Can we speak of consensual coding?

**(Reply to R2-05):** The analysis was primarily conducted by first author, and emerging themes were discussed jointly with co-authors. This is reflected in authors contributions and we have now more explicitly explained this in the Section 2.2 on data analysis.

**(R2-06):** Line 462: Interestingly despite the more physical science background, lots of the comments are on social issues and the integration of disciplines. It would be interesting to read the thoughts of the authors on this issue.

**(Reply to R2-06):** We have reflected on this in the original manuscript L420-L430. In the revised version, we also added the following new paragraph towards the end of the discussion (this is also one paragraph we use in response to R1-02, suggesting that we look at some of the reasons behind the issues/trends).

'Our findings indicate that even with a physical-science oriented set of participants, the themes of challenges in Q1 that go beyond the realm of typical physical science were deemed important (e.g., Integration of social factors, Inadequate translation of science to policy and practice, Lack of interdisciplinary approaches). These can also be attributed to the rising prominence of interdisciplinarity in natural hazards and disaster studies (Peek and Guikema, 2021). This rise has occurred as researchers have become increasingly aware that these issues are social ones as much as they are natural, and require collaboration between natural scientists, social scientists, medicine and the humanities, amongst others. Some of this concern arises from experience, in which scientists have struggled with issues of communication or recognised the role played by vulnerability in the outworking of particular crises and have extended their interests beyond single discipline science. Examples include Barclay et al. (2008) which builds on experience from multiple volcanic crises; and Bretton et al. (2015) which tackles issues arising from the L'Aquila earthquake.''

**Editor (E) [Comment sent to authors, before sending out for review]**

**(E-01):** Thank you for your manuscript. When scanning it, I asked myself if you have quantitative information on the overlap of addressees of the survey contacted through the different channels (line 105). If this is the case, it could be very interesting to include it. I guess, however, this can be done at a later stage in the review process, and is not essential before sending the manuscript to reviewers.

**(Reply to E-01):** We do not have information on the overlaps between the EGU NH (Natural Hazard) Twitter feed (1900 followers), EGU NH Facebook page (994 followers), the EGU NHESS author contact list (3085 e-mails, some no longer current) and the EGU NH division of those affiliated (1550 e-mails, of which it is a 'blind' system, so we do not know who is on the list, only EGU administrative staff are aware of this). A maximum with no overlap would be 7500 (and which includes some e-mails no longer current). A minimum would be 1900 (of the twitter followers). There is most likely overlap between these different types of communication channels. We have therefore, to give a more realistic range, changed our wording (last paragraph of Section 2.1) to now read "we estimate that the advertising reached 2000 to 4000 people".